# Experimental Comparative Study of Dynamic Behavior in Solution Phase of *C*-Tetra(phenyl)resorcin[4]arene and *C*-Tetra(phenyl)pyrogallol[4]arene

**DOI:** 10.3390/molecules25102275

**Published:** 2020-05-12

**Authors:** José Luis Casas-Hinestroza, Miguel Ángel Vela Suazo, Mauricio Maldonado Villamil

**Affiliations:** Departamento de Química, Facultad de Ciencias, Universidad Nacional de Colombia-sede Bogotá, Carrera 30 # 45-03, Bogotá 111321, Colombia; jlcasash@unal.edu.co (J.L.C.-H.); mvela@unal.edu.co (M.Á.V.S.)

**Keywords:** polyhydroxylated platform, conformational behavior, conformers, dynamic studies, dynamic boat

## Abstract

The synthesis of phenyl-resorcinarenes and pyrogallolarenes is known to produce a conformational mixture of cone and chair isomers. Depending on the synthesis conditions the composition of the conformational mixture is variable; however, the cone conformer is the greatest proportion of phenyl-resorcin[4]arenes and chair conformer of pyrogallol[4]arenes. The experimental evidence suggests that phenyl-substituted resorcinarene and pyrogallolarene exist as a dynamic boat in solution.

## 1. Introduction

Among the macromolecules there are specific groups with chemical and structural characteristics that have attracted attention in recent years in different fields of research. An interesting example is the Calixarenes family—in particular resorcinarenes and pyrogallolarenes [1]—which can be obtained by reactions between aldehydes and resorcinol or pyrogallol, respectively, the latter being the main difference between the two types of macrocycles (Scheme 1). Using different phenols for the synthesis leads to different properties; for example, resorcinarenes have two active sites on the upper rim (position 2 and hydroxyl groups), where the macromolecule can be functionalized or derivatized. A different case occurs with pyrogallolarenes, which only have hydroxyl groups on the upper rim, which restricts their derivation possibilities somewhat as compared with resorcinarenes. In both cases, the cyclocondensation reaction is performed under acid catalysis conditions [2,3,4], which has the advantage of a low cost together with a minimum of synthetic steps, in addition to a minimum variation in the synthetic design of the macrocyclic molecules with different chemical functions [5].

An interesting aspect of this type of compound is that it can adopt different structural isomers. This often depends on the reaction conditions or the type of aldehyde used and the reaction time [6]. Among the most prominent isomers are those of the cone and boat type, thanks to the fact that they are more stable and are usually obtained in greater proportion when the synthesis is carried out [7,8]. The influence of the substituent on the lower rim can influence the type of conformation that can be obtained. In this way, resorcinarenes with aliphatic substitutions on the lower rim exhibit the same cone-type isomers as the majority, while the resorcinarenes that have aromatic substitutions show cone-chair-type isomeric mixtures, in which their relationship varies depending on the synthesis conditions [9]. With respect to pyrogallolarenes, the products obtained in the synthesis with aliphatic aldehydes show a tendency for the formation of cone isomers in greater proportion. On the other hand, those synthesized from aromatic aldehydes show a preference for producing chair-type isomers [10,11]. Even so, as with resorcinarenes, there are some cases where the tendency changes or an isomeric mixture is produced, depending on the synthesis conditions [12].

The *cone* and *boat* isomers have structural properties that have been beneficial in the vast majority of applications, due to the fact that these forms have an electron-rich cavity [13], which can interact with other molecules or analytes in such a way that they can stay within these macrocycles. Another important feature is that their hydroxyl groups, both in resorcinarenes and pyrogallolarenes, have an upward orientation, which sometimes can form supramolecular assemblies through hydrogen bonds, such as capsules or aggregates. These bonds commonly established between the upper edges of resorcinarenes or pyrogallolarenes are not unique forms such as supramolecular structures—there are also assemblies where agents such as solvents [14], coordinating metals [15,16] and covalent bonds [17,18] exist. There are several applications where resorcinarenes and pyrogallolarenes have contributed to significant advances, basically due to their versatility, both in synthesis and derivatives and in their molecular structure itself. A field where they have been widely used is host-guest systems [19,20,21], since the electron-rich cavity of cone-type isomers allows different types of interactions with molecules [22] to be established, especially with tetraalkylammonium-type salts or metal cations [23,24]. This has led to different areas of applications, such as sensors [25], catalysis [26,27], heavy metal complexes for purification of water tributaries [28,29,30] and chemical separations by modification of Hight Performance Liquid Chromatography (HPLC) columns [31,32,33,34], among others.

The aforementioned applications are possible because the polyhydroxylated platforms undergo conformations and conformational interconversions in solution [35], which allows host-guest interactions. For example, the conformation of resorcinarenes can be rigidified into a cone by linking the hydroxyl groups of the upper rim, which provides a higher degree of preorganization. Nevertheless, in solution the *rccc* isomer may adopt cone and boat conformations [36,37,38], which interconvert rapidly at room temperature. Studies of the conformational properties of resorcinarenes modified on the lower rim show that the most stable conformer in solution is cone and this trend is favored by bulky substituents in the macrocyclic ring [39].

Continuing with our studies of the structure of polyhydroxylated platforms [12,33,40,41], we found that the bulky aromatic substituents on the lower rim of polyhydroxylated platforms in *cone* conformation exhibited a dynamic behavior, which stimulated interest in examining the conformational preferences. For this purpose, in this article, the comparative behavior of *C*-tetra(phenyl)-resorcin[4]arene (**1**) and *C*-tetra(phenyl)pyrogallol[4]arene (**2**) compounds is studied using ^1^H-NMR and ^13^C-NMR data and dynamic ^1^H-NMR spectra.

## 2. Results and Discussion

### 2.1. Synthesis and Separation of Conformers

As mentioned in the introduction, we chose *C*-tetra(phenyl)-resorcin[4]arene (**1**) and *C*-tetra(phenyl)pyrogallol[4]arene (**2**) in order to explore the dynamic behavior of cone conformers in dimethyl sulfoxide and acetonitrile. In this way, the obtaining of **1** and **2** was done according to the procedure described in the literature, through the acid-catalized cyclocondensation of benzaldehyde with resorcinol or pyrogallol, respectively [8,9]. The synthesis was carried out through the acid-catalyzed cyclocondensation of phenol with benzaldehyde in ethyl alcohol at reflux condition. In the reaction (Scheme 2), two products were obtained (Appendix A, which were used for dynamic nuclear magnetic resonance (NMR) studies after separation.

In the ^1^H-NMR spectrum of **1a** (Appendix A), individual assignments of the protons were made based on their positions, multiplicities, integral values and comparison of spectral data with reported values of similar compounds [12,40]. In this way, the ^1^H-NMR spectrum of **1a** displayed the characteristic signal of a methine bridge at 5.63 ppm. In the aromatic region, normally the *ortho*- and *meta*-protons of resorcinarene moiety attached to a hydroxyl group produce separate signals and the *ortho*-protons are shielded by hydroxyl groups, while the meta-protons are unshielded and resonate in the upfield region. Given this, the hydrogen of the tetrasubstituted resorcinol units appears at 6.32 and 6.15 ppm, respectively. The signals at 6.97 and 6.75 were attributed to the hydrogen in the aromatic ring of the phenyl substituent on the lower rim. Finally, the signal at 8.53 ppm was assigned to hydroxyl groups in the molecule. As mentioned above, resorcinarenes can exist in conformations of various symmetries; in this way, in the first product formed the ^1^H-NMR showed the characteristic signals for the cone conformation (diastereomer *rccc*). Initially in our case, the signals indicate the existence of highly symmetric cone conformation in solution if the ^1^H-NMR spectrum of **1a** is recorded at 333 K but if the ^1^H-NMR spectrum of **1a** is recorded at room temperature, the signals allowed inferring the presence of other conformations, as will be analyzed later. The ^13^C-NMR spectrum in DMSO-*d_6_* (Appendix A) exhibited nine signals, which agree with the structure of compound **1a**, that is, it displayed eight signals for the aromatic systems and the signal at 41.6 ppm confirmed the presence of a methyne-bridge fragment between the aromatic rings, signal assignment was confirmed using the 2D-NMR-HSQC spectrum (Appendix A).

The second product **1b**, obtained in the synthesis, exhibited absorptions for C-O stretching (1159 cm^−1^), an aromatic ring (1614 cm^−1^) and the hydroxyl groups (3318 cm^−1^) in the FT-IR spectrum (Appendix A). The ^1^H-NMR spectrum (Appendix A) displayed the characteristic signal of a methine-bridge fragment between the aromatic rings (5.56 ppm) and the aromatic hydrogen of the tetrasubstituted resorcinol unit at 6.12 and 6.22 ppm for the protons in the ortho position and the signal at 6.61 ppm for meta-protons, the signals at 6.84 were attributed to the hydrogen in the aromatic ring of the phenyl substituent on the lower rim. Finally, the signals at 8.57 and 8.46 ppm were assigned to two types of hydroxyl groups in the molecule. The signals observed in ^13^C-NMR spectrum are consistent with the structure (Appendix A).

The reaction of pyrogallol with benzaldehyde was carried out under the same conditions. After 12 h in reflux, Thin Layer Chromatography (TLC) analysis of the reaction mixture showed two products corresponding to the conformational mixture—cone (**2a**) and chair (**2b**) (Appendix A). Then the separation of *cone* and *chair* conformers was carried out by using the solvent-extraction technique.

The ^1^H-NMR spectrum of product **2a** (Appendix A) exhibited two single peaks, at 7.77 and 7.65 ppm, corresponding to two classes of hydroxyl groups, signals corresponding to the pyrogallol residues—the first signal corresponds to a hydroxyl group in position 2 and the second signal for the hydroxyl group in positions 1 and 3. Additionally, all the patterns were consistent with the structure of the expected cone conformer **2a**. On the other hand, the spectrum of product **2b** (Appendix A) exhibited four different hydroxyl moieties, at 7.86, 7.67, 7.56 and 7.45 ppm, corresponding to two classes of hydroxyl groups attached to pyrogallol residues in the macrocyclic system. Careful analysis of all the patterns confirmed the structure of chair conformer **2b**. The increase of signals in the ^13^C-NMR spectrum also confirmed the chair conformation (Appendix A). A comparison of the NMR spectra for the two isomer types is shown in Figure 1.

### 2.2. Dynamic ^1^H-NMR Studies

As mentioned above, if the ^1^H-NMR spectrum in DMSO-d_6_ of **1a** was recorded at room temperature, the signal in the aromatic region at 6.32 ppm was observed as a broad signal. This fact was interesting, because this behavior is characteristic of a dynamic system in solution. In order to establish this behavior in DMSO-d_6_ initially, the ^1^H-NMR spectrum was recorded at a temperature of 333 K (Figure 1), observing that this signal was better defined, so it was decided to perform a dynamic study at variable temperature. As shown in Figure 2, at the lowest temperature, the spectrum of Compound **1a** in the aromatic region (resorcinol residue) exhibited four signals—two for aromatic protons on the upper rim at 6.55 and 6.20 ppm and two signals for the protons on the lower rim at 6.00 and 6.15 ppm.

The ^1^H-NMR spectra of conformer **1a** at 253.15 K was consistent with the presence of two conformers that exhibited two resonances for a flattened ring and two resonances for an opposite ring. This observation prompted a detailed study of the NMR spectra. It has been well established that the conformation of the resorcinarene skeleton can be assigned in solution by a comparison of the chemical shift for the arene resonance in other analogues macrocycles, because the protons in the flattened ring are more shielded. In this way, flattened cone A and flattened cone B (Figure 3) are possible conformations in DMSO-d_6_ for this macrocyclic system, which is confirmed by other areas of the spectrum, which are consistent with this conformational assignment. In the same way, in DMSO, the spectrum obtained at 268.15 K shows coalescence of the peaks in the aromatic zone, particularly for resorcinol ring signals, confirming a very dynamic interconversion system between the conformers *flattened cone* A and *flattened cone* B.

At other temperatures (283.15, 298.15 and 333.15 K), aromatic protons in ortho- and meta-positions of the resorcinol unit are sensitive to the chemical environment of the different conformations, as has been observed in other similar systems and is favored by the interaction with the solvent. In this way, the ^1^H-NMR spectrum of **1a** in DMSO showed mixed conformations, according to the following signals—two protons in the aromatic region (6.32 and 6.15 ppm) indicate a high degree of conformational equilibrium between cone and flattened cone (A and B) conformers for diastereomer *rccc* (Figure 3).

^1^H-NMR spectrum of **2a** in DMSO-d_6_ at 293.15 K is characteristic of the cone conformation and this change when the temperature decreases (Figure 4). At 248.15 K, the spectrum showed two resonances consistent with the presence of two conformers—flattened cone A and flattened cone B (Figure 3), which showed one resonance at 6.00 ppm for a flattened pentasubstituted pyrogallol unit and one resonance at 6.60 ppm for an opposite pentasubstituted pyrogallol unit and the spectrum obtained at 273.15 K shows coalescence of these peaks in the aromatic zone. Similarly, and as seen with compound **1a**, these observations suggest that the compound exists in two forms in solution, cone and flattened cone conformers. In DMSO-d_6_, **1b** showed a similar pattern of signals and this confirms the presence of the two conformers with very small differences with respect to their coalescence temperatures.

According to the known experimental evidence for aliphatic resorcinarenes and pyrogallolarenes, those exhibit a rigid cone(*rccc*) structure at room temperature in solution and solid state [42,43] which is established due to the intramolecular hydrogen bonds in the upper rim. In contrast with the behavior of platforms with aliphatic substituent, the experimental evidence for platforms **1a** and **1b** showed that they have a dynamic behavior in solution, this behavior can be due to loss of intramolecular hydrogen bonds on the upper rim, the existence of π-π interactions of the aromatic system in the lower rim and polar··· π, C-H···π [44,45] between solvent molecules and aromatic system permitting a temporal rearrangement of the macrocycle.

## 3. Materials and Methods

Infrared (IR) spectra were recorded on a ThermoFisher Scientific Nicolet iS10 Fourier transform infrared (FTIR) spectrometer with a monolithic Diamond, Attenuated Total Reflection (ATR) accessory and absorption in cm^−1^ (Thermo Scientific, Waltham, MA, USA). ^1^H and ^13^C-NMR spectra were recorded at 400 MHz on a Bruker Advance 400 instrument. Molar mass was determined with Agilent 6470 triple quadrupole mass spectrometer. RP–HPLC analyses were performed on a Chomolith^®^ C18 column (Merck, Kenilworth, NJ, USA, 50 mm), using an Agilent 1200 Liquid Chromatograph (Agilent, Omaha, NE, USA). Chemical shifts are reported in ppm, using the solvent residual signal. Melting points were measured on a Stuart apparatus (Cole-Parmer, Stafford, UK) and are not corrected. The elemental analysis for carbon and hydrogen was carried out using a Thermo Flash 2000 elemental analyzer (Thermo Scientific, Waltham, MA, USA).

### 3.1. Synthesis of Tetra(phenyl)-pyrogallol[4]arene and Resorcin[4]arene

**Synthesis of *C*-tetra(phenyl)-pyrogallol[4]arene.** A Pyrogallol solution (5 mmol) in 10 mL of ethanol was added dropwise to 0.4 mL of concentrated chlorine acid (37%), the mixture was stirred at 0 ^º^C for 5 min and then 0.5 mL of benzaldehyde (5 mmol) was added dropwise. After 10 min, the mixture was heated to 70–80 °C and refluxed for 12 h and the solid precipitate was filtered and washed with a mixture of water and ethanol (1:1), producing a pink solid (569 mg) at 56% yield, which was characterized by means IR, ^1^H-NMR and ^13^C-NMR and used for separation experiments.

**Conformation separation.** The separation of boat and chair conformers was carried out by using the solvent-extraction technique with mixtures of solvents such as ethyl acetate, water and ethanol. The crude mixture (200 mg) was stirred in 10 mL of a mixture of solvents (ethyl acetate, ethanol and water, at a ratio of 2:1:1, respectively) for 10 min and the suspended solid was separated by filtration, dried and used for conformational studies, whereupon it was determined to be in the chair conformer configuration. The mixture of solvents was removed by evaporation, resulting in a second red solid that exhibited the boat conformation.

***C*-tetra(phenyl)pyrogallol[4]arene (*boat*)** m.p. > 350 °C IR (ATR/cm^−1^): 3300–3600 (broad), 1630, 1500, 1464, 1368, 1282, 1247, 1209, 1064, 1015, 163, 699, 566, 416 cm^−1^; ^1^H-NMR (400 MHz, DMSO-*d*_6_): δ 7.77 (s, 4H, OH); 7.65 (s, 8H, OH); 6.95–6.96 (m, 12H, Ar); 6.76–6.77 (m, 8H, Ar); 6.04 (s, 4H, Ar); 5.78 (s, 4H, CH). ^13^C-NMR (100 MHz, DMSO-*d*_6_): δ (ppm) 145.4; 142.0; 131.8; 128.6; 127.0; 124.5; 121.3. 41.3. Calcd. For C_52_H_40_O_12_ (%) C 72.89; H 4.72; O 22.41. Found: C 73.85; H 4.67; O 21.49.

***C*-tetra(phenyl)pyrogallol[4]arene (*chair*)** m.p. > 350 °C IR (ATR/cm^−1^): 3300–3600 (broad), 1630, 1500, 1464, 1368, 1282, 1247, 1209, 1064, 1015, 163, 699, 566, 416; ^1^H-NMR (400 MHz, DMSO-d_6_): δ 7.86 (s, 2H, OH); 7.67 (s, 2H, OH); 7.56 (s, 4H, OH); 7.45 (s, 4H, OH), 6.84 (s, 12H, Ar); 6.60 (*d*, *J* = 8 Hz, 8H, Ar); 6.01 (s, 2H, Ar); 5.67 (s, 4H, CH); 5.20 (s, 2H, Ar). ^13^C-NMR (100 MHz, DMSO-*d*_6_): δ (ppm) 143.8; 141.9; 141.6; 131.8; 131.3; 128.7; 126.7; 124.2; 122.4; 121.5; 121.4; 119.8; 42.6. Calcd. For C_52_H_40_O_12_ (%) C 72.89; H 4.72; O 22.41. Found: C 73.52; H 4.21; O 21.91.

**Synthesis of *C*-tetra(phenyl)-resorcin[4]arene.** A resorcinol solution (10 mmol) in 20 mL of ethanol was added to 10 mmol of benzaldehyde, the mixture was stirred at 90 °C and 2.5 mL of concentrated hydrochloric acid (37%) was added dropwise. The solution was refluxed for 6 h and the precipitated solid was filtered and washed with ethanol, producing a green solid at 86% yield. The green powder was dried and used for ^1^H-NMR analysis. Water was added to the ethanol used in the reaction and washed by inducing the precipitate. The obtained solid was filtered, washed with water and was recrystallized from acetonitrile. The second yellow solid at 14% yield was dried and used for conformational studies.

***C*-tetra(phenyl)-resorcin[4]arene(*boat*)** m.p. > 300 °C IR (KBr/cm^−1^): 3300–3600 (broad), 1600, 1470, 1440, 1380, 1340, 1210, 1260, 1150, 850, 740, 700; ^1^H-NMR(400MHz, DMSO-*d*_6_): δ 8.53 (s, 8H, OH); 6.98–6.96 (m, 12H, Ar); 6.75–6.74 (m, 8H, Ar); 6.32 (br s, 4H, CH *meta* to OH); 6.15(s, 4H, CH *orto* to OH); 5.63 (s, 4H, CH). ^13^C-NMR(100 MHz, DMSO-*d*_6_) δ (ppm): 152.4; 145.6, 130.7; 128.4; 126.8; 124.3; 120.5; 102.2; 41.6.

***C*-tetra(phenyl)-resorcin[4]arene(*chair*)** m.p. > 300 °C IR (KBr/cm^−1^): 3300–3600 (broad), 1600, 1470, 1440, 1380, 1340, 1210, 1260, 1150, 850, 740, 700; ^1^H-NMR(400MHz, DMSO-d_6_): δ 8.52 (s, 4H, OH); 8.41 (s, 4H, OH); 6,80 (s, 12H, Ar); 6.57 (s, 8H, Ar); 6.31 (s, 4H, Ar); 6.08 (s, 2H, Ar); 5.52 (s, 2H, Ar); 5.49 (s, 4H, CH). ^13^C-NMR (100 MHz, DMSO-*d*_6_) δ (ppm): 152.7; 152.5; 144.7; 131.7; 129.1; 128.9; 126.9; 124.3; 121.0; 120.6; 101.6; 42.1.

### 3.2. Dynamic Studies

The dynamic studies were carried out by using ^1^H-NMR. The solids **1a** and **2a** were dissolved in a mixture of solvents such as DMSO-*d*_6_ and acetonitrile (MeCN). The solution was cooled within a range of 296.15 K to 258.15 K taking ^1^H-NMR spectra each 5 or 10 K.

## 4. Conclusions

*C*-tetra(phenyl)-resorcin[4]arene and *C*-tetra(phenyl)-pyrogallol[4]arene were synthesized as conformational mixtures composed of the boat and the chair isomers. The analysis of the conformational mixture showed the formation of *cone* isomers in a greater proportion for the *C*-tetra(phenyl)-resorcin[4]arene, while for *C*-tetra(phenyl)-pyrogallol[4]arene, the *chair* isomer had the greater proportion. The dynamic studies in the solution phase showed that structures **1a** and **2a** are a dynamic *boat* in the solution phase, in contrast with previous studies, where the structure was reported as a *cone* conformer. The dynamic behavior is a consequence of absence of intramolecular hydrogen bonds on the upper rim, as an effect of the aromatic substituent on the lower rim.

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
