# Peer review of "Experimental Comparative Study of Dynamic Behavior in Solution Phase of C-Tetra(phenyl)resorcin[4]arene and C-Tetra(phenyl)pyrogallol[4]arene"

_molecules, 2020, doi:10.3390/molecules25102275_

Round 1

Reviewer 1 Report

The manuscript descibes the synthesis of phenyl-resorcinarenes and pyrogallolarenes and show that depending on the substituents there is a preference to synthesize cone or chair conformations of compounds. The authors conduct dynamic NMR studies which reveal that the compounds are in a boat conformation in contrast to previous studies. The manuscript is very good written with clear describtions of all the experiments.

The only my question is why the authors have not used 2D NMR methods to clarify the exact conformations of the receptors. Is it possible?

Have the authors tried to crystalize the molecules?

Are there any other methods to prove the conformation in solution? This would help to answer a number of questions.

Author Response

We are sending the corrected manuscript Molecules-756967 title “Experimental Comparative study of Dynamic Behavior in Solution Phase of C-tetra(phenyl)resorcin[4]arene and C-tetra(phenyl)pyrogallol[4]arene” by José Luis Casas-Hinestroza. Miguel Angel Vela Suazo and Mauricio Maldonado. It was modified according with the Reviewers comments.

Following you will find, in blue, the modifications made in the manuscript as well as the answers to each Reviewer comments.

The only my question is why the authors have not used 2D NMR methods to clarify the exact conformations of the receptors. Is it possible?

Answer: We agree with the reviewer, unfortunately our NMR has some limitations to make these determinations at variable temperature

Have the authors tried to crystalize the molecules?

Answer: We agree with the reviewer; several attempts were made but its crystallization was not possible.

Are there any other methods to prove the conformation in solution? This would help to answer a number of questions.

Answer: We agree with the reviewer, it is possible to derivatize these products to evaluate their conformations by NMR spectroscopy.

Reviewer 2 Report

This manuscript describes a comparative conformational study of tetra(phenyl)-resorcin[4]arene and tetra(phenyl)pyrogallol[4]arene compounds in solution by 1H and 13C NMR experiments, as well as VT-NMR studies. Although this work seems to have been done with care, the results are not of enough significance to warrant publication in Molecules. Moreover, there are some typographical errors and the English language needs improvement. More care should be paid to the manuscript writing.

Thus, in my opinion, this manuscript is not suitable for publication in Molecules.

Some specific comments:

- Page 1, line 5: is the corresponding author name correct? (“Suazoand”?)

- Page 1, line 25: it should be (“Scheme 1”)

- Page 1, line 27: “(position 2 the hydroxyls and hydroxyl groups)” ??

- Page 1, line 29: “which only have hydroxyl groups on the upper rim”. Re-write the sentence, as resorcinarenes also have only hydroxyl groups on the upper rim.

- Page 2, lines 55 and 58: it should be “resorcinarenes” and “unique forms”

- Page 3, line 80: the title of this part cannot be only “Results”, as part 3 is also results.

- Page 3, line 87: it should be “Scheme 2”

- Page 3, lines 86-91: the synthesis details should be moved to the “Materials and Methods” part (point 4.1)

- Page 3, lines 94-112: these 1H and 13C NMR spectra should be placed in SI and it should be mentioned here as Figure S1, and so on.

- Page 4, lines 124-125: solvent details should be removed

- Page 4, Fig. 1 caption: solvent and temperature conditions are missing

- Page 4, line 128: it should be “spectrum”; what do authors mean by “first product”?

- Page 5, line 145: it should be Figure 2?

- Page 5, Figure 2 caption: solvent and compound number are missing; line 150 (Fig. 2): from which compounds are these NMR spectra?

- Page 5, line 156: it should be Figure 3

- Page 5: all section 3 is very confusing

- Page 6, line 170: it should be Figure 3

- Page 6, line 172: it should be Figure 4

- Page 6, line 174: it should be Figure 3

- Page 8, line 248: it should be “296.15 K”

- Page 8, lines 250 and 253: “C-tetra(phenyl)-pyrogallol[4]arene” is repeated

- Page 8, line 256: “The dynamic……” unclear sentence

Author Response

Dear Reviewer,

We are sending the corrected manuscript Molecules-756967 title “Experimental Comparative study of Dynamic Behavior in Solution Phase of C-tetra(phenyl)resorcin[4]arene and C-tetra(phenyl)pyrogallol[4]arene” by José Luis Casas-Hinestroza. Miguel Angel Vela Suazo and Mauricio Maldonado. It was modified according with the Reviewers comments.

Following you will find, in blue, the modifications made in the manuscript as well as the answers to each Reviewer comments.

This manuscript describes a comparative conformational study of tetra(phenyl)-resorcin[4]arene and tetra(phenyl)pyrogallol[4]arene compounds in solution by 1H and 13C NMR experiments, as well as VT-NMR studies. Although this work seems to have been done with care, the results are not of enough significance to warrant publication in Molecules. Moreover, there are some typographical errors and the English language needs improvement. More care should be paid to the manuscript writing.

Thus, in my opinion, this manuscript is not suitable for publication in Molecules.

Some specific comments:
- Page 1, line 5: is the corresponding author name correct? (“Suazoand”?) Answer: We agree with the reviewer. It was corrected (line 5)

Page 1, line 25: it should be (“Scheme 1”)

Answer: We agree with the reviewer. It was corrected (line 25)

Page 1, line 27: “(position 2 the hydroxyls and hydroxyl groups)” ??

Answer: We agree with the reviewer. It was corrected (lines 27)

Page 1, line 29: “which only have hydroxyl groups on the upper rim”. Re-write the sentence, as resorcinarenes also have only hydroxyl groups on the upper rim.

Answer: The correction on line 27 gives meaning to the sentence “which only have hydroxyl groups on the upper rim”

 Page 2, lines 55 and 58: it should be “resorcinarenes” and “unique forms” Answer: We agree with the reviewer. It was corrected (lines 55 and 58)

Page 3, line 80: the title of this part cannot be only “Results”, as part 3 is also results.

Answer: We agree with the reviewer. It was corrected (line 139)

Page 3, line 87: it should be “Scheme 2”

Answer: We agree with the reviewer. It was corrected (line 87)

Page 3, lines 86-91: the synthesis details should be moved to the “Materials and Methods” part (point 4.1)

Answer: We agree with the reviewer. It was corrected (lines 87 and 88) and was changed for “and the two products obtained as indicated in the experimental part were used in the NMR studies”.

Page 3, lines 94-112: these 1H and 13C NMR spectra should be placed in SI and it should be mentioned here as Figure S1, and so on.

Answer: We agree with the reviewer. We include the support information (SI)

Page 4, lines 124-125: solvent details should be removed

Answer: We agree with the reviewer. It was corrected (lines 122 - 122)

Page 4, Fig. 1 caption: solvent and temperature conditions are missing

Answer: We agree with the reviewer. It was corrected (line 124)

Page 4, line 128: it should be “spectrum”; what do authors mean by “first product”?

Answer: We agree with the reviewer. It was corrected (line 125)

Page 5, line 145: it should be Figure 2?

Answer: We agree with the reviewer. It was corrected (line 142).

Page 5, Figure 2 caption: solvent and compound number are missing; line 150 (Fig. 2): from which compounds are these NMR spectra?

Answer: We agree with the reviewer. It was corrected (line 147).

Page 5, line 156: it should be Figure 3 Answer: We agree with the reviewer. It was corrected (line 143).
Page 5: all section 3 is very confusing.

Answer: We agree with the reviewer, however with the suggested changes it is not confusing.

Page 6, line 170: it should be Figure 3

Answer: We agree with the reviewer. It was corrected (line 167).

Page 6, line 172: it should be Figure 4

Answer: We agree with the reviewer. It was corrected (line 169).

Page 6, line 174: it should be Figure 3

Answer: We agree with the reviewer. It was corrected (line 171).

Page 8, line 248: it should be “296.15 K”

Answer: We agree with the reviewer. It was corrected (line 245).

Page 8, lines 250 and 253: “C-tetra(phenyl)-pyrogallol[4]arene” is repeated Answer: We agree with the reviewer. It was corrected (lines 247 and 250).

Page 8, line 256: “The dynamic……” unclear sentence

Answer: We agree with the reviewer. It was corrected (lines 253 - 254).

Reviewer 3 Report

Review report_ molecules-756967-peer-review-v1

The manuscript “ Experimental Comparative study of Dynamic Behavior in Solution Phase of C-tetra(phenyl)resorcin[4]arene and C-tetra(phenyl)pyrogallol[4]arene" by Casas-Hinestroza, Vela and Maldonado present the synthesis of crown and chair conformation of phenyl-resorcinarenes and pyrogallolarenes. The main finding in this work is the separation of the cone and chair conformer of the phenyl-resorcinarenes and pyrogallolarenes.

Atwood and co-workers in 2016 published the work “Process development for separation of conformers from derivatives of resorcin[4]arenes and pyrogallol[4]arenes” Chem. Eur. J. 2016, 22, 43, 15202-15207. https://doi.org/10.1002/chem.201603090

It was cited herein as reference 6.

Atwood and co-workers in their work prepared 4-hydroxyphenylresorcin[4]arenes and 4-hydroxyphenylpyrogallol[4]arene. They separated using the same processes, isolated them, solve the crystal structures, etc. This work is the same. I see no advancement herein. Based on that published work by Atwood and coworkers, I find this work unpublishable, unfortunately.

Author Response

Dear Reviewer,

We are sending the corrected manuscript Molecules-756967 title “Experimental Comparative study of Dynamic Behavior in Solution Phase of C-tetra(phenyl)resorcin[4]arene and C-tetra(phenyl)pyrogallol[4]arene” by José Luis Casas-Hinestroza. Miguel Angel Vela Suazo and Mauricio Maldonado. It was modified according with the Reviewers comments.
Following you will find, in blue, the modifications made in the manuscript as well as the answers to each Reviewer comments.

The manuscript “ Experimental Comparative study of Dynamic Behavior in Solution Phase of C-tetra(phenyl)resorcin[4]arene and C-tetra(phenyl)pyrogallol[4]arene" by Casas-Hinestroza, Vela and Maldonado present the synthesis of crown and chair conformation of phenyl-resorcinarenes and pyrogallolarenes. The main finding in this work is the separation of the cone and chair conformer of the phenyl-resorcinarenes and pyrogallolarenes.

Atwood and co-workers in 2016 published the work “Process development for separation of conformers from derivatives of resorcin[4]arenes and pyrogallol[4]arenes” Chem. Eur. J. 2016, 22, 43, 15202-15207. https://doi.org/10.1002/chem.201603090

It was cited herein as reference 6.
Atwood and co-workers in their work prepared 4-hydroxyphenylresorcin[4]arenes and 4-hydroxyphenylpyrogallol[4]arene. They separated using the same processes, isolated them, solve the crystal structures, etc. This work is the same. I see no advancement herein. Based on that published work by Atwood and coworkers, I find this work unpublishable, unfortunately.

Answer: As mentioned by the reviewer, the manuscript of Atwood and co-workers describes the synthesis, the separation of the conformational mixture of 4-hidroxyphenyl-resorcin[4]arene and 4-hydroxyphenylpyrogallol[4]arene and show the DRX studies of the products. According to the above, we want to highlight the following aspects of our manuscript:

- The manuscript of Atwood and co-coworkers focuses on a solid state study, whereas in our case it was done in solution, the principles approach can be considered quite different.
- We conduct dynamic NMR studies in solution which reveal that the compounds are in a boat conformation and show the conformational equilibrium of the studied macrocycles in contrast to previous studies. Therefore, these conclusions are not feasible with the results presented by Atwood and co-coworkers.
- According to our bibliographic review, this is the first time that dynamic behavior has been evidenced in solution for resorcinarenes and pirogalolarenes substituted on the lower rim with aromatic rings.
- Our study allows us to explain the lack of clarity of NMR signals in some cases.
- Finally, the products obtained by Atwood and co-workers were obtained with p-hydroxybenzaldehyde while we start from benzaldehyde, the presence of the 2 hydroxyl group completely changes the behavior of these derivatives in solution.

In this way, we consider that our manuscript will be interesting to the readership of Molecules and serve as a good model for NMR-guided study of macrocycle conformations.

Round 2

Reviewer 2 Report

General comments:

The above manuscript has been improved, but still needs substantial corrections.

Specific comments:

- Page 2, line 55: it should be “unique forms”

- Page 3, line 80: the title of this part cannot be only “Results”; It should be “Results and Discussion” and it needs to have a sub-title as “Synthesis and conformational analysis”, for example.

- Page 3, lines 86-88: it should be as ”In the reaction    (Scheme 2), two products were obtained (Figure S1), which were used    studies after separation.”

- Page 3, line 91: It should be: “spectrum of 1a (Figure S2)”

- Page 3, line 106: It should be: “The 13C spectrum (Figure S3)”; Authors should speak about the HSQC experiment they did (Figure S4) and also about the IR (Figure S5). The SI Figures should be numbered consecutively and introduced just after be mentioned; for example: line 111 “in the FT-IR spectrum (Figure S6); line 112 “The 1H NMR spectrum (Figure S7)”. And so on.

- Page 4, lines 122: the sentence should end at “solvent-extraction technique.”

- Page 4, lines 126, 130: it should be “spectrum of product 2a” and “the spectrum of product          2b”

- Page 5, Figure 2 caption: it should be “resorcin[4]arene” and the compound number are still missing.

- Page 6, Figure 4 caption: the solvent and the compound number are missing. Moreover, in all the NMR Figures the magnetic field is missing too (is it 400 MHz?)

- SI document: Scheme S1 is the same as Scheme 2 (on page 3 of the manuscript), so, it   should be removed from SI; the same for Figure S6 (the same as Figure 2).

- Figure S10 should be mentioned in the manuscript.

- Figure S16 is the same as S12 (IR spectrum of 2a).

Author Response

General comments:

The above manuscript has been improved, but still needs substantial corrections.

Specific comments:

Page 2, line 55: it should be “unique forms

Answer: We agree with the reviewer. It was corrected (line 55)

Page 3, line 80: the title of this part cannot be only “Results”; It should be “Results and Discussion” and it needs to have a sub-title as “Synthesis and conformational analysis”, for example.

Answer: We agree with the reviewer. It was corrected (lines 80-81)

Page 3, lines 86-88: it should be as ”In the reaction (Scheme 2), two products were obtained (Figure S1), which were used studies after separation.”

Answer: We agree with the reviewer. It was corrected (lines 87-89)

Page 3, line 91: It should be: “spectrum of 1a (Figure S2)”

Answer: We agree with the reviewer. It was corrected (line 92)

Page 3, line 106: It should be: “The 13C spectrum (Figure S3)”; Authors should speak about the HSQC experiment they did (Figure S4) and also about the IR (Figure S5). The SI Figures should be numbered consecutively and introduced just after be mentioned; for example: line 111 “in the FT-IR spectrum (Figure S6); line 112 “The 1H NMR spectrum (Figure S7)”. And so on.

Answer: We agree with the reviewer. It was corrected (lines 88, 92, 107, 109, 110, 113, 122)

Page 4, lines 122: the sentence should end at “solvent-extraction technique.”

Answer: We agree with the reviewer. It was corrected (line 124)

Page 4, lines 126, 130: it should be “spectrum of product 2a” and “the spectrum of product 2b”

Answer: We agree with the reviewer. It was corrected (lines 127 and 131)

Page 5, Figure 2 caption: it should be “resorcin[4]arene” and the compound number are still missing.

Answer: We agree with the reviewer. It was corrected (line 149)

Page 6, Figure 4 caption: the solvent and the compound number are missing.
Moreover, in all the NMR Figures the magnetic field is missing too (is it 400 MHz?)

Answer: We agree with the reviewer. It was corrected (line 181)

SI document: Scheme S1 is the same as Scheme 2 (on page 3 of the manuscript), so, it should be removed from SI; the same for Figure S6 (the same as Figure 2).

Answer: We agree with the reviewer. It was corrected

Figure S10 should be mentioned in the manuscript.

Answer: We agree with the reviewer. It was corrected (line 88)

Figure S16 is the same as S12 (IR spectrum of 2a)

Answer: We agree with the reviewer. It was corrected

Reviewer 3 Report

The authors did a good job of addressing the weaknesses and explaining the clear differences between their work and that of Atwood et al. Chem. Eur. J. 2016, 22, 43, 15202-15207. https://doi.org/10.1002/chem.201603090

I think the work is now ready to move to the next level

Author Response

The authors did a good job of addressing the weaknesses and explaining the clear differences between their work and that of Atwood et al. Chem. Eur. J. 2016, 22, 43, 15202-15207. https://doi.org/10.1002/chem.201603090

I think the work is now ready to move to the next level

Answer: We agree with the reviewer